# Sodium Butyrate Inhibits the Expression of Thymidylate Synthase and Induces Cell Death in Colorectal Cancer Cells

**DOI:** 10.3390/ijms25031572

**Published:** 2024-01-26

**Authors:** Nayeon Kim, Changwon Yang

**Affiliations:** Department of Science Education, Ewha Womans University, Seoul 03760, Republic of Korea; nayeon7112@ewhain.net

**Keywords:** colorectal cancer, sodium butyrate, thymidylate synthase, apoptosis, short-chain fatty acids

## Abstract

The most commonly used chemotherapy for colorectal cancer (CRC) is the application of 5-fluorouracil (5-FU). Inhibition of thymidylate synthase (TYMS) expression appears to be a promising strategy to overcome the decreased sensitivity to 5-FU caused by high expression of TYMS, which can be induced by 5-FU treatment. Several compounds have been shown to potentially inhibit the expression of TYMS, but it is unclear whether short-chain fatty acids (SCFAs), which are naturally produced by bacteria in the human intestine, can regulate the expression of TYMS. Sodium butyrate (NaB) is the most widely known SCFA for its beneficial effects. Therefore, we investigated the enhancing effects on inhibition of cell viability and induction of apoptosis after co-treatment of NaB with 5-FU in two CRC cell lines, HCT116 and LoVo. This study suggests that the effect of NaB in improving therapeutic sensitivity to 5-FU in CRC cells may result from a mechanism that strongly inhibits the expression of TYMS. This study also shows that NaB inhibits the migration of CRC cells and can cause cell cycle arrest in the G2/M phase. These results suggest that NaB could be developed as a potential therapeutic adjuvant to improve the therapeutic effect of 5-FU in CRC.

## 1. Introduction

Colorectal cancer (CRC) is the second most common cause of cancer death in the United States and is rapidly shifting to diagnosis at a younger age at a more advanced stage [1]. 5-fluorouracil (5-FU) is a synthetic fluorinated pyrimidine analog most widely used for chemotherapy of CRC [2]. However, many CRC patients develop resistance to 5-FU, which can lead to low therapeutic sensitivity [3]. Therefore, several strategies have been considered to alleviate resistance based on the intracellular activity of 5-FU [4]. The primary effect of 5-FU is by inhibiting the activity of thymidylate synthase (TYMS), and the expression level of TYMS has an inverse relationship with sensitivity to 5-FU and is considered a potential indicator of resistance to 5-FU in CRC [5]. TYMS is an enzyme responsible for the biosynthesis of thymidylate and plays an essential role in DNA replication and repair. TYMS catalyzes the conversion of deoxyuridine monophosphate (dUMP) to deoxythymidine monophosphate (dTMP), and fluorodeoxyuridine monophosphate (FdUMP), an activated metabolite of 5-FU, binds to the nucleotide binding site of TYMS. Low TYMS in tissues from CRC patients correlated with a good prognosis in patients treated with 5-FU chemotherapy [6]. In CRC cells, treatment with 5-FU causes a rapid increase in the expression of TYMS, and repeated exposure to 5-FU confers 5-FU resistance to the cells with an increase in TYMS [7,8]. Therefore, the development of compounds that can potentially inhibit the expression of TYMS is considered a novel strategy to improve the efficiency of 5-FU-based CRC treatment.

Short-chain fatty acids (SCFAs) are produced by fermentation of indigestible dietary fiber by intestinal bacteria. Among the SCFAs produced in the intestines, sodium acetate (NaAc), sodium propionate (NaPc), and sodium butyrate (NaB) account for the majority, and their beneficial effects on the intestines have been widely reported [9,10]. For instance, SCFAs alleviate the inflammatory response induced by 5-FU and help maintain the integrity of intestinal tight junction [11]. Moreover, SCFAs produced in the human intestine can provide energy for the proliferation of intestinal epithelial cells, regulate cell metabolism, and inhibit the growth of harmful bacteria [12]. In the human colonic lumen, SCFAs are typically present at high levels in the order of mM. Although NaB has the lowest percentage among the three types of SCFAs in colonic lumen, NaB is the preferential compound for energy supply in colonocytes, whose oxidation satisfies at least 60% of energy needs [13]. Furthermore, a number of published data reveal that NaB is the most associated SCFA type in preventing the progression of CRC [14]. One line of evidence shows that NaB induces downregulation of thioredoxin-1 (Trx-1) in CRC cells, increasing reactive oxygen species (ROS) levels and inhibiting cell growth [15]. In another study, NaB induced autophagy by dose-dependently increasing the phosphorylation of AMP-activated protein kinase (AMPK) in CRC cells [16]. As such, the signaling pathways and molecular targets regulated by NaB in colon cancer cells are being revealed, but little research has been carried out regarding the effect of NaB on TYMS. Moreover, it is unclear whether NaB can improve the therapeutic effect when combined with 5-FU, a conventional anticancer drug.

A previous study showed that a derivative of butyrate inhibits the mRNA level expression of TYMS and may have a synergistic effect with 5-FU on apoptosis in CRC, suggesting that NaB may serve as a potential therapeutic adjuvant for 5-FU [17]. Therefore, in this study, the efficacy of NaB for regulating the protein level expression of TYMS was analyzed in CRC cells. Since 0 to 10 mM is generally recognized as the recommended treatment concentration of NaB in in vitro models, we treated CRC cells with NaB in a dose-dependent manner within this range. After that, we investigated whether NaB could regulate cell viability, apoptosis, and cell cycle, and then confirmed whether the therapeutic effect on CRC was improved through combined treatment with 5-FU. 

## 2. Results

### 2.1. NaB Impaired CRC Cell Viability and Migration

We first verified whether NaB could inhibit the viability of colon cancer cells. NaB (0, 0.1, 0.2, 0.5, 1, 2, 5, and 10 mM) dose-dependently inhibited the viability of HCT116 and LoVo cells (Figure 1A). NaB significantly inhibited cell viability in HCT116 when exposed to concentrations above 1 mM, while it inhibited cell viability in LoVo when exposed to concentrations above 2 mM. Therefore, we set 2 mM as the maximum NaB treatment concentration in subsequent experiments. Moreover, the proportion of viable HCT116 and LoVo cells decreased upon NaB treatment in a dose-dependent manner, as shown by crystal violet staining results (Figure 1B). An amount of 2 mM NaB reduced viable cell numbers by 71.3% in HCT116 cells and 79.3% in LoVo cells. Migration between cells was also inhibited after NaB treatment. Furthermore, we analyzed whether NaB caused migration-inhibiting effects in CRC cells (Figure 1C). Through experiments based on wound healing, we showed that treatment with 2 mM NaB for 12 h significantly reduced migration in HCT116 and LoVo cells. These results suggest that NaB clearly has the effect of inhibiting the viability and migration of CRC cells.

### 2.2. NaB Induced CRC Cell Apoptosis

Next, we analyzed whether NaB affected the apoptosis of CRC cells. We found that the addition of NaB (0.5, 1, and 2 mM) increased the apoptosis rate (8.8%, 12.1%, 16.5%) of HCT116 cells compared with cells not treated (1.6%) (Figure 2A). Similarly, the addition of NaB (0.5, 1, and 2 mM) increased the apoptosis rate (8.6%, 11.5%, 17.6%) of LoVo cells compared with cells not treated (6.7%). The cleavage of poly (ADP-ribose) polymerase (PARP) is an important characteristic of DNA damage in cancer cells [18]. NaB significantly increased the expression of cleaved PARP in HCT116 and LoVo cells (Figure 2B). These results suggest that NaB may cause CRC cell death.

### 2.3. NaB in Combination with 5-FU Enhanced CRC Cell Growth Suppression and Apoptosis Induction

Next, we analyzed whether NaB could enhance the effect when combined with 5-FU, a common anticancer drug for CRC. When applied alone for 48 h, 5-FU (0, 1, 2, 5, 10, 20, 40, and 80 μM) dose-dependently inhibited the viability of HCT116 and LoVo cells (Figure 3A). We confirmed that the addition of NaB (2 mM) combined with 5-FU induced greater reductions in the viability of HCT116 and LoVo cells than 5-FU treatment alone. Crystal violet staining provided evidence that NaB could further enhance the effect of 5-FU-induced reduction in cell viability (Figure 3B). Combination treatment of NaB with 5-FU significantly reduced the number of viable HCT116 and LoVo cells compared with treatment with 5-FU alone. Furthermore, addition of NaB more effectively inhibited spheroid formation in HCT116 and LoVo cells compared with exposure to 5-FU alone (Figure 3C). These results suggest that the addition of NaB may have a stronger inhibitory effect on cell viability than when treated with 5-FU alone in CRC cells.

Moreover, Annexin V and propidium iodide (PI) staining revealed that 5-FU significantly induced apoptosis of HCT116 and LoVo cells (Figure 4A). The addition of NaB increased the apoptosis rate (8.5%) of HCT116 cells compared with 5-FU treatment alone (4.0%). Similarly, the addition of NaB increased the apoptosis rate (24.3%) of LoVo cells compared with 5-FU treatment alone (20.5%), although was not statistically significant. It was confirmed by analyzing the expression of cleaved PARP that combined treatment with NaB enhanced apoptosis with 5-FU (Figure 4B). Treatment with 5-FU alone slightly induced PARP cleavage in HCT116 and LoVo cells, whereas addition of NaB caused significantly higher PARP cleavage. These results suggest that NaB may be effective as an anticancer adjuvant that can be applied to CRC together with 5-FU.

### 2.4. NaB Decreased TYMS Expression in CRC Cells

We next analyzed changes in the expression of TYMS after treating CRC cells with NaB to determine whether TYMS was involved in the effect of NaB in enhancing the effect of 5-FU. As a result, NaB (2 mM) clearly reduced the mRNA level of *TYMS* in HCT116 and LoVo cells (Figure 5A). In contrast, the mRNA levels of methylenetetrahydrofolate reductase (MTHFR) and dihydropyrimidine dehydrogenase (DPYD), which are involved in the intracellular metabolism of 5-FU, increased upon NaB treatment in HCT116 and LoVo cells (Figure 5B,C). Analysis of the protein level expression of TYMS showed that NaB significantly inhibited the expression of TYMS in both HCT116 and LoVo cells (Figure 5D). As previously shown, treatment of 5-FU in CRC cells dramatically increased the expression of TYMS (Figure 5E). In particular, there was a clear increase in the expression of the upper band, suggestive of bound TYMS. In HCT116 and LoVo cells, addition of NaB significantly alleviated the 5-FU-induced increase in TYMS. These results were similar for the upper band, suggesting bound TYMS, as well as the lower band, suggesting unbound TYMS. These results suggest that NaB can inhibit the expression of TYMS at both mRNA and protein levels in CRC cells. It also suggests that the effect of NaB in suppressing the expression of TYMS may be one of the potential molecular mechanisms to improve the effect of 5-FU in CRC.

### 2.5. The Higher Concentration of NaB Further Shifted the Cell Cycle Stage Distribution of CRC Cells toward G2/M

We next analyzed whether NaB could regulate cell cycle distribution in CRC cells by measuring changes in DNA content. As shown in Figure 6, NaB (0, 0.5, 1, and 2 mM) dose-dependently decreased the ratio of G1 phase while relatively increasing the ratio of G2/M phase. In HCT116 cells, exposure to 0.5, 1, and 2 mM NaB caused 41.6%, 40.5%, and 49.7% of cells to remain in the G2/M phase, respectively, which was significantly higher than the 23.4% in the control group. Similarly, in LoVo cells, exposure to 0.5, 1, and 2 mM NaB resulted in 34.1%, 35.5%, and 46.5% of cells entering the G2/M phase, respectively, compared with 26.1% in the control group. These results suggest that NaB may cause G2/M arrest in CRC cells.

### 2.6. NaB Alone and in Combination with 5-FU Enhanced 5-FU Resistant CRC Cell Growth Suppression and Apoptosis Induction

We repeatedly treated 5-FU to HCT116 cells to make them resistant to 5-FU and assessed the effect of NaB on 5-FU-resistant HCT116 (HCT116-5-FUR) cells. In HCT116-5-FUR cells, 5-FU did not significantly inhibit cell viability at concentrations up to 20 μM, suggesting that they were less sensitive to 5-FU compared with parental HCT116 cells (Figure 7A). On the other hand, 2 mM NaB treatment also led to a significant decrease in cell viability in HCT116-5-FUR cells in the presence or absence of 5-FU, as in HCT116 cells. Similarly, 5-FU (20 μM) did not lead to a significant increase in apoptosis in HCT116-5-FUR cells, but the addition of NaB clearly enhanced the proportion of apoptotic cells (Figure 7B). These results were supported by the finding that the cleavage of PARP was increased in response to NaB treatment, unlike 5-FU treatment, in HCT116-5-FUR cells (Figure 7C). Meanwhile, the protein level expression of TYMS showed that the upper band was observed even in the control group, indicating that bound TYMS was present at a higher level in HCT116-5-FUR cells compared with parental HCT116 cells (Figure 7D). The expression of TYMS was significantly increased upon 5-FU treatment in HCT116-5-FUR cells as in parental HCT116 cells. In HCT116-5-FUR cells, addition of NaB reduced the expression level of TYMS compared with treatment with 5-FU alone. Although treatment with NaB alone did not inhibit the expression of TYMS compared with the control, these results suggest that NaB may inhibit the upregulation of TYMS, which can be increased by 5-FU. Taken together, the additional application of NaB can be considered as one of the ways to effectively inhibit cell viability and induce apoptosis in CRC cells resistant to 5-FU.

## 3. Discussion

This study verified that NaB can reduce cell viability and induce apoptosis in CRC cells within the concentration range commonly used in an in vitro model [19]. In particular, the addition of NaB in CRC cells significantly improved the sensitivity to 5-FU compared with treatment with 5-FU alone. Epidemiological studies show that about half of CRC cases are preventable through appropriate nutrition and diet [20]. A lower incidence of CRC is observed in populations consuming high proportions of dietary fiber, suggesting an involvement of dietary fiber in CRC risk [21]. SCFAs are present in concentrations ranging from 50 to 200 mM in the human large intestine and are the main metabolites of aerobic fermentation by microbial communities. Butyrate is a major SCFA produced when indigestible dietary fiber in foods is fermented and decomposed by intestinal bacteria. In humans, butyrate accounts for the smallest production proportion of approximately 15% among the three representative SCFAs (acetate, propionate, and butyrate), but its beneficial effects on intestinal metabolism are attracting attention [22]. In particular, butyrate is considered to be the major energy source for approximately 70% of the energy needs of the colonic mucosa [23]. It has been reported that chronic NaB supplementation in food helps alleviate several metabolic and physiological abnormalities in the body. Although it has been shown that oral administration of butyrate can alleviate mucositis induced by 5-FU, it has been unclear whether butyrate can improve the anticancer efficacy of 5-FU in CRC [24]. This study clearly demonstrated that NaB could further enhance the cell death of CRC cells when combined with 5-FU. This suggests that, as previously known, NaB not only contributes to alleviating cytotoxicity and inflammation that may appear in the intestines due to the administration of 5-FU but also improves the anticancer efficacy of 5-FU.

Several lines of evidence have suggested that NaB reduces the survival rate of CRC cells. At a concentration range of up to 5 mM, NaB induces a dose-dependent decrease in viability and an increase in apoptosis of HT29 and SW480 cells [15,25]. Another study shows that NaB at concentrations up to 32 mM can inhibit the growth of HCT116 cells in a dose- and time-dependent manner [26]. Other evidence shows that NaB also promotes the cleavage of PARP in HCT116 and HT29 cells, similar to our findings [27,28]. Together with previous studies, our findings clearly demonstrate the potential therapeutic effect of NaB on CRC. In addition, our study suggests that NaB can further improve the anticancer effect of 5-FU, especially by regulating the expression of proteins targeted by 5-FU. Contrary to our results, one study shows that NaB does not exhibit a synergistic effect with 5-FU in CRC cells, although it further inhibits cell growth in response to combined treatment with oxaliplatin, another common anticancer drug [26]. Therefore, the effect of NaB on improving therapeutic sensitivity to standard anticancer drugs for CRC needs to be studied closely in the future to resolve this controversy.

5-FU is converted to FdUMP in cells and blocks the activity of TYMS, preventing the conversion of dUMP to dTMP. Cellular properties that can be disrupted by 5-FU are extensive, including apoptosis, cell cycle, glucose metabolism, oxidative stress, mitochondrial function, and epithelial–mesenchymal transition [2]. Several lines of evidence have suggested that the administration of adjuvants that can downregulate TYMS may promote the anticancer effect of 5-FU. Our previous study showed that apigenin, a plant-derived compound, inhibits the expression of TYMS in HCT116 and HT29 cells and increases sensitivity to 5-FU [29]. In CRC cells consistently exposed to 5-FU, 5-FU resistance is established with high intracellular TYMS expression, as shown in our results [5]. The upper band of TYMS in the Western blot images indicates the expression of FdUMP-conjugated TYMS, which shows a rapid increase rate upon exposure of 5-FU in cells. What causes the resistance of CRC cells to 5-FU is that the expression of unbound TYMS is also increased when exposed to 5-FU, as implied by the increased intensity of the lower band in Western blot images. In other words, if the activity of TYMS is inhibited by the binding of FdUMP in CRC cells, the expression of TYMS will increase at the mRNA level by negative feedback. Therefore, improving the sensitivity of 5-FU through inhibition of TYMS requires fundamentally suppressing gene expression, and we show that NaB can perform this function well in CRC cells. The effect of treatment with NaB on inhibiting the expression of TYMS at the mRNA and protein levels is comparable to that of any plant-derived compounds reported in our previous studies [29]. These results suggest that NaB is a promising compound to inhibit gene expression of TYMS and may be of sufficient value as an adjuvant to improve the anticancer effect of 5-FU in CRC.

Our study also suggests that NaB can cause cell cycle arrest and inhibit proliferation in colon cancer cells. In particular, it was clearly confirmed that colon cancer cells undergo G2/M arrest in response to NaB. In addition, NaB appears to effectively inhibit the migration of CRC cells, suggesting that the effects of NaB on CRC may be multifaceted. A closer investigation of the molecular mechanism of NaB needs to be conducted in future studies. Meanwhile, our results suggest that NaB alone does not inhibit the expression of TYMS in 5-FU-resistant CRC cells. Recent evidence shows that butyrate-insensitive properties can be acquired in HCT116-5-FUR cells [30]. Although it is unclear whether NaB can regulate the expression of TYMS in CRC resistant to 5-FU, our study suggests that it may inhibit the expression of TYMS, which may be increased by 5-FU during treatment.

MTHFR and DPYD are enzymes that play important roles in the metabolic pathway of 5-FU-based chemotherapy. MTHFR converts 5,10-methylenetetrahydrofolate (5,10-MTHF) to 5-methylenetetrahydrofolate (5-MTHF) [31]. 5-MTHF is the dominant form of folate in plasma and can provide a methyl group during the biosynthesis of various metabolites [32]. Moreover, 5,10-MTHF forms a ternary complex with 5-FU and TYMS and can inhibit methylation from dUMP to dTMP, stably maintaining the effect of 5-FU. Meanwhile, DPYD functions as a rate-limiting enzyme in the intracellular catabolism of 5-FU, converting 5-FU to 5-fluoro-5,6-dihydrouacil [33]. Previous reports have shown that, similar to the expression of TYMS, low levels of DPYD expression have been implicated in eliciting a sensitive response to 5-FU [34]. Therefore, the increase in mRNA level expression of MTHFR and DPYD by NaB shown in our results may have a negative effect on improving the sensitivity to 5-FU. Future studies need to analyze how regulating the expression of MTHFR and DPYD in CRC cells affects the intracellular metabolism of 5-FU.

Taken together, our study is, to the best of our knowledge, the first report that NaB can regulate the expression of TYMS in CRC cells at the mRNA and protein levels. Furthermore, the high therapeutic sensitivity improvement effect of NaB when combined with 5-FU suggests the value of NaB as a potential therapeutic adjuvant. As inhibition of TYMS expression is emerging as a promising molecular strategy to improve the therapeutic sensitivity of 5-FU, it is expected to increase the treatment efficiency of CRC through the production of NaB, which is naturally made from dietary fiber by gut microbiota.

## 4. Materials and Methods

### 4.1. Cell Culture

The human CRC-derived cell lines HCT116 and LOVO were obtained from the Korean Cell Line Bank (Seoul, Republic of Korea). Both cells were cultured in 37 °C and 5% CO_2_ using RPMI-1640 medium containing 10% fetal bovine serum. HCT116-5-FUR were established by treatment with gradually increasing concentrations of 5-FU for approximately 6 months. At each subculture, dead cells or debris floating in the supernatant were removed, and only viable cells were harvested. NaB (303410) and 5-FU (F6627) was purchased from Sigma-Aldrich (St. Louis, MO, USA).

### 4.2. Cell Viability Test

Experiments were carried out using the Cell Proliferation Kit I (MTT) (11465007001, Roche, Basel, Switzerland) according to the manufacturer’s instructions. The cells were dispensed into 96-well plates and incubated for 24 h, and then treated with 5-FU and NaB. After 48 h of incubation, 3-(4,5-dimethylthiazol-2-yl)-2,5-diphenyltetrazolium bromide (MTT) labeling agent was treated and incubated at 37 °C for 4 h, and then a solubilization solution was dispensed and incubated at 37 °C overnight. It was then quantified by measuring absorbance at 590 nm and 650 nm using a Multiskan SkyHigh Microplate Spectrophotometer (Thermo Fisher Scientific, Waltham, MA, USA).

### 4.3. Crystal Violet Staining

Cells were cultured in 24-well plates and then treated with 20 μM of 5-FU, 2 mM of NaB, or their combination for 48 h. The medium was removed and the cells were fixed with cold methanol for 10 min. Then, the treated cells were stained by adding 0.5% crystal violet solution (V5265, Sigma-Aldrich) for 10 min. After this, the cells were washed several times with phosphate buffered saline, and images of the stained cells were acquired.

### 4.4. Migration Assay

Cells were allowed to adhere to the well and grow to greater than 90% confluence in Culture-Insert 2 Well in µ-Dish (80206, ibidi GmbH, Gräfelfing, Germany). Then, cells were treated with 2 mM NaB, the wall between cells was removed, and cultured for 12 h. The changes in the wound area were monitored under a CKX53 phase contrast microscope (Olympus, Tokyo, Japan). The migration distance of each group was analyzed using ImageJ software comparing the area observed at 0 h.

### 4.5. Annexin V and PI Staining

Cells were allowed to adhere to the well and grow to greater than 90% confluence. Apoptosis of CRC cells was analyzed using the Fluorescein isothiocyanate Annexin V apoptosis detection kit I (BD Biosciences, Franklin Lakes, NJ, USA). CRC cells were treated with 20 μM of 5-FU, 2 mM of NaB, or their combination for 48 h. Then, cells were harvested and washed several times with phosphate buffered saline. The cell suspension was incubated with FITC Annexin V and PI for 15 min at room temperature in the dark. The fluorescence intensity of the samples was analyzed using a LSRFortessa™ (BD Biosciences) at Ewha Fluorescence Core Imaging Center.

### 4.6. Western Blot Analysis

Western blotting was performed to analyze the expression levels of the proteins extracted from CRC cells. Proteins were extracted using RIPA lysis buffer, and protein concentration was determined using Bradford protein assay (Bio-Rad, Hercules, CA, USA) with bovine serum albumin as a standard. Proteins were separated by SDS-PAGE, transferred to nitrocellulose membranes, and exposed overnight to a primary antibody. Then, the membrane was exposed to peroxidase-conjugated secondary antibody for 1 h, and band images were obtained using KwikQuant Pro Imager (Kindle Biosciences, Greenwich, CT, USA). α-tubulin (TUBA) antibody was used to normalize the intensity of band images. All primary antibodies used were purchased from Cell Signaling Technology (Danvers, MA, USA).

### 4.7. Spheroid Culture

CRC spheroids were prepared based on the hanging drop method. Briefly, cells diluted in growth medium to a concentration of 1 × 10^5^/mL were dropped into the lid of a culture dish. The bottom of the culture dish was filled with phosphate buffed saline, which served as the hydration chamber. The lids were inverted onto the bottom chamber and incubated cells at 37 °C and 5% CO_2_ until cell sheets or aggregates formed. Cells were treated with 20 μM of 5-FU, 2 mM of NaB, or their combination for 3 days. Changes in spheroid morphology were observed using a CKX53 phase contrast microscope.

### 4.8. Quantitative Polymerase Chain Reaction (qPCR)

To quantify gene expression of NaB-treated cells, total RNA was extracted from cells using AccuPrep^®®^ Universal RNA Extraction Kit (K-3140, Bioneer, Daejeon, Republic of Korea). Complementary DNA was synthesized using AccuPower^®®^ RT PreMix (K-2261, Bioneer). Then gene expression was quantified using primer pairs for *TYMS* (forward: 5′-CCA AAG CTC AGG ATT CTT CG-3′, reverse: 5′-AGT TGG ATG CGG ATT GTA CC-3′), *MTHFR* (forward: 5′-CTT TGA GGC TGA CCT GAA GC-3′, revesre: 5′-AAG CGG AAG AAT GTG TCA GC-3′) and *DPYD* (forward: 5′-AAT GAC TGG ACG GAA CTT GC-3′, reverse: 5′-CAT TCC TCT TTC TCC CAT GC-3′). For qPCR, AccuPower^®®^ 2X GreenStar™ qPCR Master Mix (K-6252, Bioneer) was used according to the manufacturer’s instructions. Relative mRNA levels were calculated using the 2^−ΔΔCT^ method based on C_T_ values. The expression of the *GAPDH* gene was measured for normalization.

### 4.9. Cell Cycle Assay

CRC cells were treated with 20 μM of 5-FU, 2 mM of NaB, or their combination for 48 h. Then, cells were harvested and washed several times with phosphate buffered saline. Cells were fixed overnight in 70% ethanol and washed again with phosphate buffered saline several times. Next, cells were treated with RNase A and PI for 30 min, and the fluorescence intensity of the samples were analyzed using a LSRFortessa™ at Ewha Fluorescence Core Imaging Center.

### 4.10. Statistical Analysis

All experiments were independently repeated three times. The statistical significance of each experiment was verified using SAS statistical analysis software (version 9.4).

## Figures and Tables

**Figure 1 ijms-25-01572-f001:**
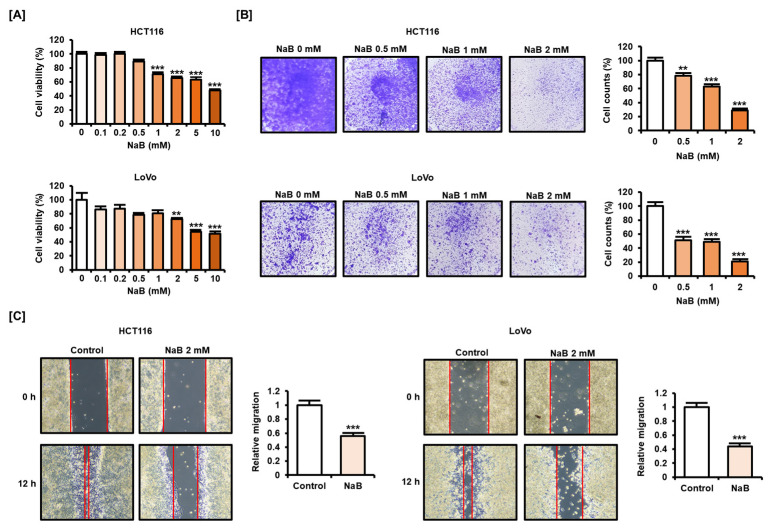
Effect of sodium butyrate (NaB) on the viability and migration of colorectal cancer (CRC) cells. (**A**) Cell viability of HCT116 and LoVo cells was verified after dose-dependent treatment with NaB (0, 0.1, 0.2, 0.5, 1, 2, 5, and 10 mM) for 48 h. (**B**) After HCT116 and LoVo cells were dose-dependently treated with NaB (0, 0.5, 1, and 2 mM) for 48 h, visualization of living cells was performed using crystal violet staining. The stained cells were washed with phosphate buffered saline for taking macrographic images. The number of stained cells was analyzed via ImageJ software (version 1.53e). (**C**) Changes in migration of HCT116 and LoVo cells following NaB (2 mM) treatment for 12 h were analyzed through the approximate distance between left and right cell populations. Photographs were taken under phase contrast microscope at 40× magnification. All experiments were performed in three independent replicates, and the statistical significance of the results is indicated by asterisks (** *p* < 0.01 and *** *p* < 0.001).

**Figure 2 ijms-25-01572-f002:**
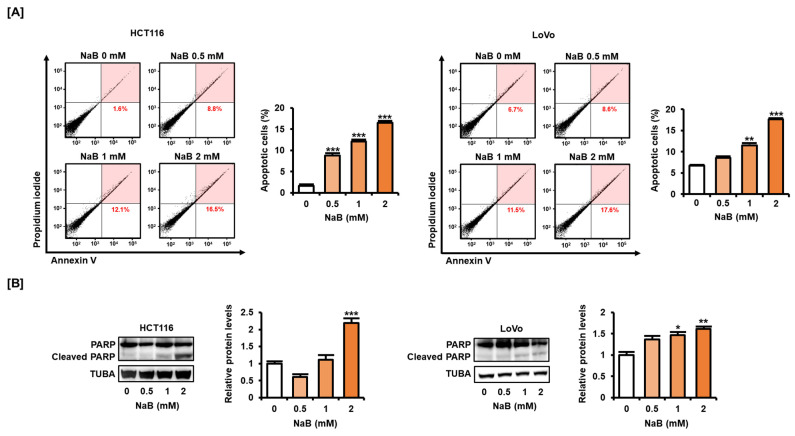
Effect of NaB on the apoptosis of CRC cells. (**A**) Changes in apoptosis in HCT116 and LoVo cells after treatment with NaB (0, 0.5, 1, and 2 mM) in a dose-dependent manner for 48 h were performed by Annexin V and propidium iodide (PI) staining. The upper right part of the four quadrants of the dot plot obtained using a flow cytometer was quantified and displayed as a graph. (**B**) Protein expression of poly (ADP-ribose) polymerase (PARP) and cleaved PARP was analyzed by Western blot. Proteins were extracted from HCT116 and LoVo cells treated with NaB (0, 0.5, 1, and 2 mM) in a dose-dependent manner for 24 h. Expression of α-tubulin (TUBA) was used for normalization. All experiments were performed in three independent replicates, and the statistical significance of the results is indicated by asterisks (* *p* < 0.05, ** *p* < 0.01, and *** *p* < 0.001).

**Figure 3 ijms-25-01572-f003:**
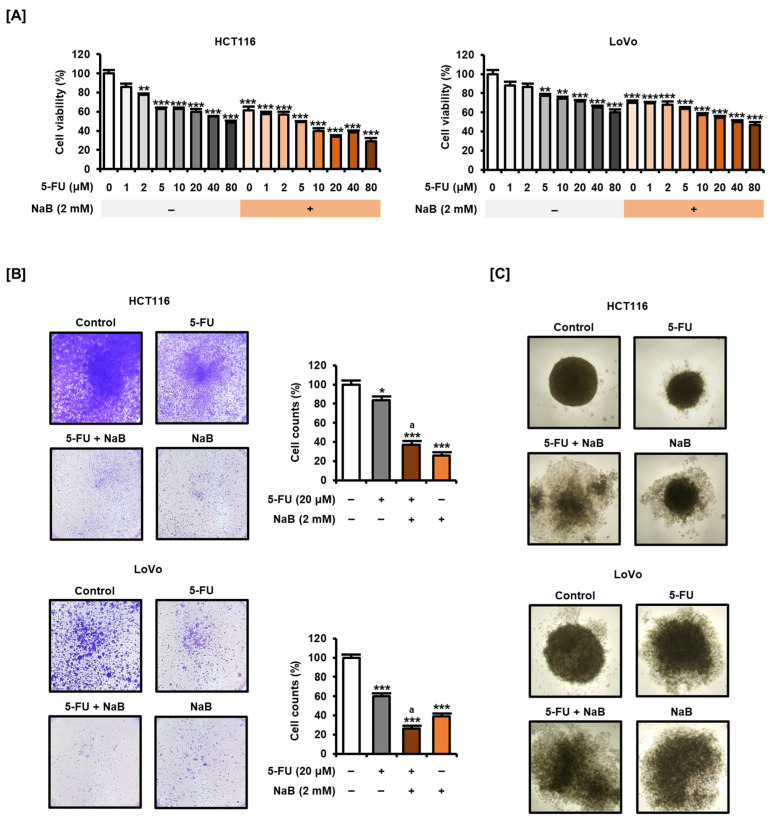
Effect of combined treatment of 5-fluorouracil (5-FU) and NaB on the viability of CRC cells. (**A**) Cell viability of HCT116 and LoVo cells was verified after dose-dependent treatment with 5-FU (0, 1, 2, 5, 10, 20, 40, and 80 μM) for 48 h depending on the presence or absence of NaB (2 mM). (**B**) After HCT116 and LoVo cells were dose-dependently treated with 5-FU (20 μM) and NaB (2 mM) for 48 h, visualization of living cells was performed using crystal violet staining. The number of stained cells was analyzed via ImageJ software. (**C**) Spheroid formation of HCT116 and LoVo cells was imaged after treatment with 5-FU (20 μM) and NaB (2 mM) for 3 days. Photographs were taken under phase contrast microscope at 40× magnification. All experiments were performed in three independent replicates, and the statistical significance of the results is indicated by asterisks (* *p* < 0.05, ** *p* < 0.01, and *** *p* < 0.001). ‘a’ means there is statistical significance (*p* < 0.05) of combined treatment of 5-FU and NaB compared with 5-FU treatment alone.

**Figure 4 ijms-25-01572-f004:**
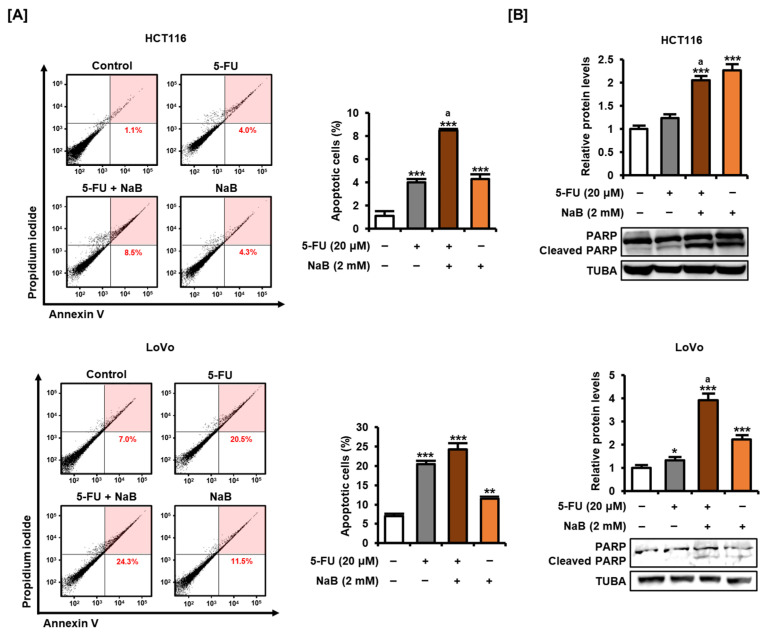
Effect of combined treatment of 5-FU and NaB on the apoptosis of CRC cells. (**A**) Changes in apoptosis in HCT116 and LoVo cells after treatment with 5-FU (20 μM) and NaB (2 mM) for 48 h were performed by Annexin V and PI staining. The upper right part of the four quadrants of the dot plot obtained using a flow cytometer was quantified and displayed as a graph. (**B**) Protein expression of PARP and cleaved PARP was analyzed by Western blot. Proteins were extracted from HCT116 and LoVo cells treated with 5-FU (20 μM) and NaB (2 mM) for 24 h. Expression of TUBA was used for normalization. All experiments were performed in three independent replicates, and the statistical significance of the results is indicated by asterisks (* *p* < 0.05, ** *p* < 0.01, and *** *p* < 0.001). ‘a’ means there is statistical significance (*p* < 0.05) of combined treatment of 5-FU and NaB compared with 5-FU treatment alone.

**Figure 5 ijms-25-01572-f005:**
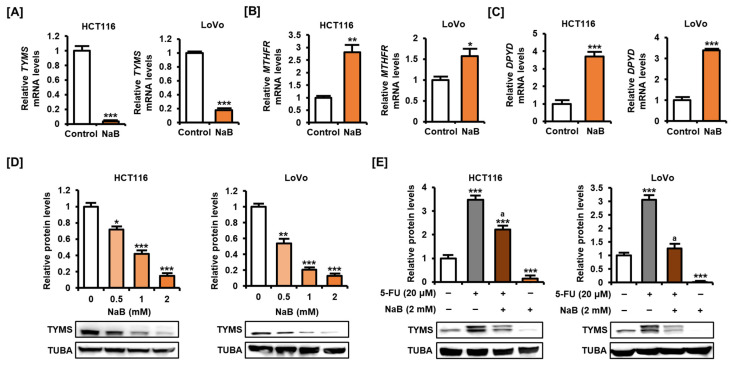
Effect of NaB on TYMS expression in CRC cells (**A**–**C**). The mRNA level expression of thymidylate synthase (TYMS) (**A**), methylenetetrahydrofolate reductase (MTHFR) (**B**), and dihydropyrimidine dehydrogenase (DPYD) (**C**) was analyzed via qPCR. Total RNA was extracted after treating HCT116 and LoVo cells with NaB (2 mM) for 24 h. (**D**) Protein expression of TYMS was analyzed by Western blot. Proteins were extracted from HCT116 and LoVo cells treated with NaB (0, 0.5, 1, and 2 mM) for 24 h. (**E**) Protein expression of TYMS was analyzed by Western blot. Proteins were extracted from HCT116 and LoVo cells treated with 5-FU (20 μM) and NaB (2 mM) for 24 h. Expression of TUBA was used for normalization. All experiments were performed in three independent replicates, and the statistical significance of the results is indicated by asterisks (* *p* < 0.05, ** *p* < 0.01, and *** *p* < 0.001). ‘a’ means there is statistical significance (*p* < 0.05) of combined treatment of 5-FU and NaB compared with 5-FU treatment alone.

**Figure 6 ijms-25-01572-f006:**
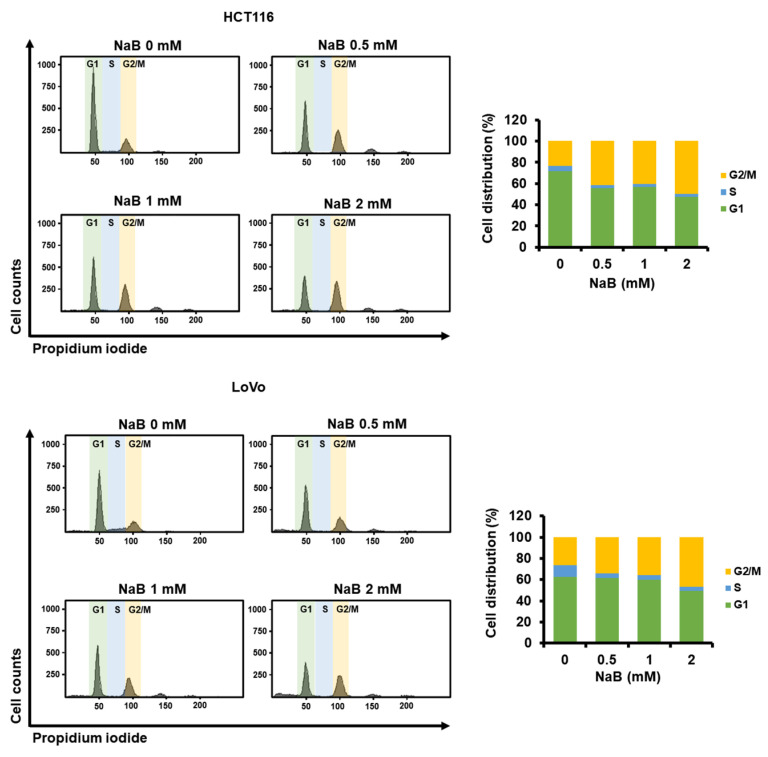
The effect of NaB on cell cycle distribution in CRC cells was analyzed through PI staining. HCT116 and LoVo cells were treated with NaB (0, 0.5, 1, and 2 mM) in a dose-dependent manner. Cells were classified into G1, S, and G2/M phases according to the amount of DNA in the cells using flow cytometry. All experiments were performed in three independent replicates.

**Figure 7 ijms-25-01572-f007:**
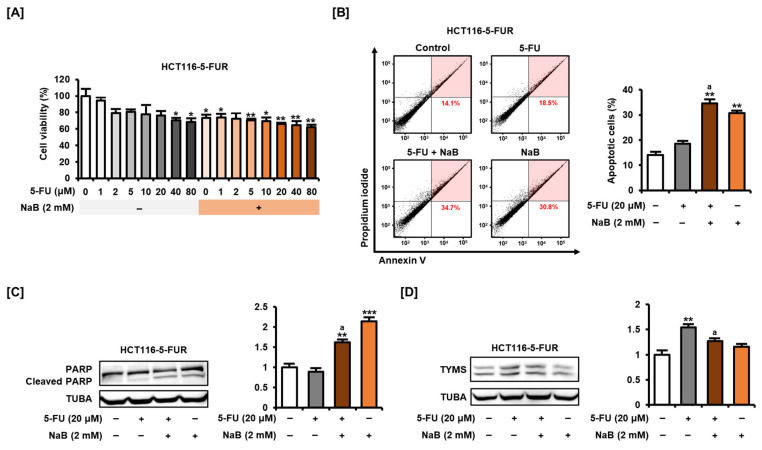
Effect of combined treatment of 5-FU and NaB in 5-FU-resistant HCT116 (HCT116-5-FUR) cells. (**A**) Cell viability of HCT116-5-FUR cells was verified after dose-dependent treatment with 5-FU (0, 1, 2, 5, 10, 20, 40, and 80 μM) for 48 h depending on the presence or absence of NaB (2 mM). (**B**) Changes in apoptosis in HCT116-5-FUR cells after treatment with 5-FU (20 μM) and NaB (2 mM) for 48 h were performed by Annexin V and PI staining. The upper right part of the four quadrants of the dot plot obtained using a flow cytometer was quantified and displayed as a graph. (**C**) Protein expression of PARP and cleaved PARP was analyzed by Western blot. Proteins were extracted from HCT116-5-FUR cells treated with 5-FU (20 μM) and NaB (2 mM) for 24 h. (**D**) Protein expression of TYMS was analyzed by Western blot. Proteins were extracted from HCT116-5-FUR cells treated with 5-FU (20 μM) and NaB (2 mM) for 24 h. Expression of TUBA was used for normalization. All experiments were performed in three independent replicates, and the statistical significance of the results is indicated by asterisks (* *p* < 0.05, ** *p* < 0.01, and *** *p* < 0.001). ‘a’ means there is statistical significance (*p* < 0.05) of combined treatment of 5-FU and NaB compared with 5-FU treatment alone.

## Data Availability

The authors will share the data upon request.

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
