# Peer review of "Sodium Butyrate Inhibits the Expression of Thymidylate Synthase and Induces Cell Death in Colorectal Cancer Cells"

_ijms, 2024, doi:10.3390/ijms25031572_

Round 1
Reviewer 1 Report
Comments and Suggestions for Authors
In this manuscript, the authors investigate possible therapeutic effects of Sodium butyrate alone and combined with 5-FU treatment on colorectal cancer by using two human CRC cell lines. This paper is interesting and could be of interest to the CRC patients. I have some mayor and minor comments about this study that are numbered below:
1. Please clarify all the abbreviations (like TYMS, SCFA, etc..) in the beginning of the manuscript when the abbreviation appears for the first time (preferably introduction).
2. How are the concentrations of NAB and 5-FU selected? Did the authors have any prior knowledge of the suggested concentrations in these or normal colon cells? Is their effect on normal colon cells known?
3. How was the housekeeping gene for Western blot and qPCR determined? There is an ongoing discussion about the upregulation of normalization genes in CRC and some authors (like Oncotarget 2016. DOI: 10.18632/oncotarget.11439) even suggest a total protein as a housekeeping. What is the author’s opinion about this?
4. The authors perform cell viability and apoptosis tests for a 48h treatment and select the most appropriate dosage, but the spheroid formation is evaluated after 3 days treatment and protein extraction after 1-day treatment. The authors don’t have the viability and apoptosis data for a 1-day nor 3-day treatment. Why didn’t the authors perform everything for a 3-day treatment if these cells need 3 days to form spheroids? Depending on the type of drug, 24h more or 24h less could change cell viability and apoptosis.
5. In the Discussion, the authors discuss in depth their findings, but they don’t compare it to any other investigation done by other authors. Could the authors check is there any other publication with similar topic/drugs/effects and comment how similar or opposite those results are comparing with their own.
6. The authors state in the Discussion that “NaB in the biological concentration range can reduce……” (line 233). Can they explain why they consider used concentration as biological.
Reviewer 2 Report
Comments and Suggestions for Authors
In this study, the authors investigated the enhancing effects on inhibition of cell viability and induction of apoptosis after co-treatment of NaB with 5-FU in two CRC cell lines, finding that the effect of NaB in improving therapeutic sensitivity to 5-FU, by inhibiting the expression of TYMS. Although the conclusions were supported by some data, a few issues need to be addressed.
1. The background is not strong enough and some the information is not complete.
2. Please make a list for abbreviations.
3. According to the cell viability of HCT116 in Figure 1A (~20% cell death), crystal violet staining showed much more cell death (~70%) when treated with 2mM NaB?
4. For Annexin V/PI staining in Figure 2A, there was no Annexin V positive signal at all?
5. In Figure 3B, there was no big differences between NaB and NaB+5-FU group, the conclusion in Line 117-118 cannot be supported?
Comments on the Quality of English LanguageEnglish language is fine.
Round 2
Reviewer 1 Report
Comments and Suggestions for Authors
I thank the authors for addressing all my concerns and making changes to improve their manuscript, which I consider sufficiently improved to be published.
Reviewer 2 Report
Comments and Suggestions for Authors
In this study, the authors investigated the enhancing effects on inhibition of cell viability and induction of apoptosis after co-treatment of NaB with 5-FU in two CRC cell lines, finding that the effect of NaB in improving therapeutic sensitivity to 5-FU, by inhibiting the expression of TYMS. The conclusions were supported by solid data, but the sygernistic effects were not strong the authors may need to try to other combinations in the future.